# Study of CdS/CdS Nanoparticles Thin Films Deposited by Soft Chemistry for Optoelectronic Applications

**DOI:** 10.3390/mi14061168

**Published:** 2023-05-31

**Authors:** Laura Aislinn Carrasco-Chavez, José F. Rubio-Valle, Abimael Jiménez-Pérez, José E. Martín-Alfonso, Amanda Carrillo-Castillo

**Affiliations:** 1Institute of Engineering and Technology, Autonomous University of Ciudad Juarez, Juarez Chihuahua 32310, Mexico; al187107@alumnos.uacj.mx (L.A.C.-C.); abimael.jimenez@uacj.mx (A.J.-P.); 2Chemical Product and Process Technology Research Center (Pro2TecS), Department of Chemical Engineering and Materials Science, University of Huelva, 21071 Huelva, Spain; josefernando.rubio@diq.uhu.es

**Keywords:** cadmium sulfide, chalcogenide semiconductor, thin films, nanoparticles

## Abstract

Chalcogenides semiconductors are currently being studied as active layers in the development of electronic devices in the field of applied technology. In the present paper, cadmium sulfide (CdS) thin films containing nanoparticles of the same material as the active layer were produced and analyzed for their application in fabricating optoelectronic devices. CdS thin films and CdS nanoparticles were obtained via soft chemistry at low temperatures. The CdS thin film was deposited via chemical bath deposition (CBD); the CdS nanoparticles were synthesized via the precipitation method. The construction of a homojunction was completed by incorporating CdS nanoparticles on CdS thin films deposited via CBD. CdS nanoparticles were deposited using the spin coating technique, and the effect of thermal annealing on the deposited films was investigated. In the modified thin films with nanoparticles, a transmittance of about 70% and a band gap between 2.12 eV and 2.35 eV were obtained. The two characteristic phonons of the CdS were observed via Raman spectroscopy, and the CdS thin films/CdS nanoparticles showed a hexagonal and cubic crystalline structure with average crystallite size of 21.3–28.4 nm, where hexagonal is the most stable for optoelectronic applications, with roughness less than 5 nm, indicating that CdS is relatively smooth, uniform and highly compact. In addition, the characteristic curves of current-voltage for as-deposited and annealed thin films showed that the metal-CdS with the CdS nanoparticle interface exhibits ohmic behavior.

## 1. Introduction

In the last decade, chalcogenide semiconductors have been investigated for applications in optoelectronic devices due to their efficiency and diverse properties, which have great importance in the field of applied technology [1,2,3,4]. Cadmium sulfide (CdS) has been one of the most studied chalcogenide materials since Reynolds et al. observed the photovoltaic effect on CdS crystals with metallic electrodes [5,6]. In this way, CdS is an II-IV (binary) type *n* semiconductor with excellent properties, such as a band gap of 2.42 eV at room temperature, with high carrier concentration (10^16^–10^18^ cm^−^^3^) and mobility (0.1–10 cm^2^ V^−^^1^ s^−^^1^), a high absorption coefficient (>10^4^ cm^−1^), and high electrochemical stability [7,8,9]. These properties have led to its effective development for photovoltaic solar cells [10], light-emitting diodes [5], photoelectric devices [11], chemical sensors [12], surface acoustic wave devices, thin film transistors (TFTs) [6,7,13], photocatalysis and biological sensors, optical coding, optical data storage and sensing, non-linear integrated optical devices [5,14,15] and photocatalytic hydrogen generation [16].

For CdS applications mentioned above, CdS can be presented in different materials, mainly studied through thin films and nanoparticles. There are several physical and chemical methods for the deposition of CdS thin films, the most commonly used among them are vacuum evaporation [17], pulsed laser ablation (PLA) [18], electrodeposition [19], molecular beam epitaxy (MBE) [17], sputtering, successive ionic layer adsorption and reaction (SILAR) [20], chemical spray pyrolysis (CSP) [21], RF magnetron sputtering [22], hydrothermal synthesis [23], and chemical bath deposition (CBD) [8,13,24,25].

In addition, there are different processes used for the synthesis of nanoparticles, such as hydrothermal or solvothermal techniques [26,27], microwave approaches [28], and chemical vapor, but these treatments are expensive, extensive, toxic and energy-consuming. However, methods such as suspension colloidal synthesis routes, sol–gel techniques [12] and chemical precipitation processes are available at room temperature and low cost [28]. Every deposition method has its advantages and limitations; however, the soft chemistry techniques have been studied extensively due to their simplicity, low deposition temperatures, low cost, large area deposition, and low risk synthesis [2,6,8,10,12,19,20,21,29,30]. Two major and important benefits of the soft chemistry techniques can be considered: the preparation of new metastable phases and the preparation of nanosize materials.

Chemical precipitation is the most widely used method for producing nanoparticles owing to its ease, simplicity, and short reaction time [31,32]. Chalcogenide nanoparticles have attracted increasing attention from scientists in recent years due to their properties resulting from the quantum confinement effect [33]. This is because the physical properties of a bulk material are independent of its shape and size, whereas the chemical and physical properties of a nanoscale structure are influenced by its shape and size [34]. The development of thin films of chalcogenide homojunctions for the improvement of final properties, based on binary or ternary materials of the same or different chemical nature, has been reported in multiple studies [19,23,35,36,37,38]. 

This work reports the first fabrication of a homojunction based on CdS/CdS NpS using soft chemistry methods, such as chemical bath deposition for CdS thin films and the precipitation method for CdS nanoparticles. The overall objective of this work is to explore the influence of the incorporation of CdS nanoparticles into CdS thin films to improve their properties for optoelectronic applications. To this end, the optical, chemical, morphological and electrical properties of these films are investigated herein.

## 2. Materials and Methods

### 2.1. Materials

The following reagents were used for the preparation of the CdS thin films: cadmium chloride was employed as the metal source (CdCl_2_, purity ≥ 99.0%), sodium citrate (Na_3_C_6_H_5_O_7_, Fermont, purity ≥ 99.8%) and potassium hydroxide (KOH, Fermont, purity ≥ 87.3%) were used as complexing agents, thiourea was used as an anionic source (CH_4_N_2_S, J.T. Baker purity ≥ 99.4%), and finally, borate pH 10 (pH 10, J.T. Baker) was used as a pH stabilizer buffer.

Furthermore, the following reagents were used for the synthesis of CdS nanoparticles: cadmium chloride (CdCl_2_, purity ≥ 99.0%) as the metallic precursor, thioacetamide (C_2_H_5_NS, purity ≥ 99.0%) as the anionic precursor, which was provided by Merck Sigma Aldrich S.A, and sodium dodecyl sulfate (SDS, J.T. Baker, purity ≥ 95.0%) as the surfactant.

Cadmium is considered a toxic metal, and a suspected carcinogen according to the safety data sheet of the salt used in this work. Cadmium exists in an inorganic state as CdS and its toxicity is higher than that of CdS. However, CdS can be hazardous to the environment and human health due to its toxicity. the Cd precursor and CdS must be handled and used with extreme caution.

### 2.2. Deposition of CdS Thin Films

The CdS films were deposited on glass slide substrates (soda lime glass) previously washed with acetone, isopropanol, and deionized water for 10 min in each solvent sequentially under sonication (Branson 5800). For the deposition of cadmium sulfide by CBD, the deposition system was prepared as shown in Figure 1. The CdS films were deposited by immersing the glass substrates in a CBD solution according to Palma-Soto et al. [39], which was prepared from cadmium chloride (CdCl_2_), sodium citrate (Na_3_C_6_H_5_O_7_), borate buffer at a pH of 10, potassium hydroxide (KOH), and thiourea (SC(NH_2_)_2_) in a volume ratio of 9 mL (0.05 M): 9 mL (0.5 M): 3 mL: 3 mL (0.5 M): 4.5 mL (0.5 M). The total reaction volume was adjusted to 60 mL with water. For the thin film deposition on glass substrates, the temperature of the solution was 43 °C ± 1 °C for 33 min. According to multiple studies [40,41,42,43,44], the mechanisms involved in the growth of CdS films through CBD immersion, depending on the deposition time, begins with the adsorption of complex ions and the subsequent ion exchange ends the CdS formation. Each chemical bath deposit represents a layer in the thin film. Depending on the application of the optoelectronic device, thin or thick semiconductor films are required. Both thin and thick chalcogenide films with useful properties have been reported as having been deposited in CBD, where the deposition time is an important parameter related to concentration or thickness of the films [43]. In the continuous dip approach, the substrate remains in the chemical reaction bath while reactants are periodically replenished [44], and in the multiple dip approach, the substrate is repeatedly immersed in a fresh chemical solution [45,46]. To study the effects of varying concentrations of CdS in this research, thin films were made of one and two layers, which will be identified as 1 CBD CdS and 2 CBD CdS, respectively. 

### 2.3. CdS Nanoparticles (NpS) Synthesis

The precipitation method was used for the synthesis of CdS nanoparticles according to the procedures reported by Sonker et al. and Zhou et al. [12,47] at a low concentration of SDS corresponding to 0.0025 M. This was performed by depositing 50 mL of sodium dodecyl sulphate (SDS) at 0.0025 M on a CORNING PC-6200 stirring plate at 300 rpm for 5 min. Next, 10 mL of cadmium chloride (0.25 M) was added drop by drop and stirred at 300 rpm for 10 min. Finally, 20 mL of thioacetamide (0.25 M) was added drop by drop and stirred at 300 rpm for 30 min. The entire procedure is shown in Figure 2. The nanoparticles were separated by centrifugation and washed in water. The morphology reported in Rondiya et al. and Zhou et al. [6,47] for CdS NpS was spherical with hexagonal phases and a band gap of 3 eV (see Appendix A for UV-Vis absorption spectrum). Some byproducts were identified by Fourier Transform Infrared Spectroscopy (FTIR), resulting from the hydrolysis of the sulfur source and SDS used as anionic surfactants in the CdS NpS synthesis. However, the byproducts identified do not affect the optical and electrical properties of CdS nanoparticles. For these reasons, the characteristics of the CdS nanoparticles synthesized in Zhou et al. [47] make them suitable candidates for electronic technology. 

In addition, some works have reported the synthesis of CdS nanoparticles via soft chemistry processes with potential applications of electronic devices, where the stability of the materials obtained depends on the chemical or physical parameters of the synthesis [48,49]. 

### 2.4. Formation of CdS/CdS NpS Thin Films

The CdS nanoparticles were deposited on the CdS thin films. The CdS deposited on glass substrates were placed in the spin coater (anticorrosion 6” wafer Max) and were completely covered with 1 mL of CdS-as synthesized nanoparticles dispersed in isopropanol (IPA). The concentration of CdS in solution was 10 wt%. The spinning speed was set at 1000 rpm for 60 s. The drying step was performed after the CdS nanoparticle deposition at a temperature of 70 °C for 10 min in an Accuplate hot plate (Labnet International, Edison, NJ, USA) in air atmosphere. The final heat treatment was performed in a tubular oven. The samples were placed on an aluminum base in the oven at 150 °C for 30 min in air atmosphere. The identification of samples is shown in Table 1.

### 2.5. Characterization of Films: CdS by CBD and CdS by CBD/CdS NpS by Spin Coating

All prepared CdS thin films with and without CdS nanoparticles were characterized. UV-Vis Jenway 6850 was used to study the optical properties in the range of 300 nm to 1100 nm with a scan of 0.2 nm. The chemical properties were studied via Raman spectroscopy using Renishaw in Via Raman Microscope equipment in the range of 100–1000 cm^−^^1^. The morphology was studied in a SEM JEOL JSM—7000F with an operating voltage of 15 kV. For the surface analysis of the samples, we used an atomic force microscope (AFM) MFP3D-SA brand ASYLUM RESEARCH with x,y scan up to 90 µm and Z scan up to 15 µm. The structural characterization was performed by an X-ray Diffractometer Bruker Model D8 Advance with CuKα (λ) = 1.54 Å, operated at 40 kV, 30 mA and a scanning speed of 2ϴ at 0.5°/min. Chrome contacts were deposited by sputtering to determine the electrical characteristics through current-voltage measurements. This characterization was made at room temperature using the semiconductor parameter analyzer B1500A.

## 3. Results and Discussions

### 3.1. UV-Vis Spectroscopy Analysis 

Figure 3 and Figure 4 show the transmittance and absorbance spectra of the thin films. The spectra show a well-defined absorption peak at 484 nm, which is significantly blue-shifted relative to the absorption peak of bulk CdS, indicating a quantum size effect [6]. The Tauc method was used to calculate the direct band gap value from the absorption spectra. From the Tauc method:(1)$$\frac{d^{2}\tau }{dE^{2}}\equiv 0,$$
assuming a direct band gap:(2)$$\tau ={\left[\left(A)(E\right)\right]}^{2}=C\left(E-E_{g}\right)$$
where *τ* = Tauc variable, *C* = slope of linear behavior, *E_g_* = Energy band gap, *E* = Incident energy, and *A* = *A*(*E*) = Absorption of the coating.
(3)$$\frac{d^{2}\tau }{dE^{2}}=\frac{d^{2}C\left(E-E_{g}\right)\;}{dE^{2}}\equiv 0$$

In Figure 5, the geometric behavior of the direct Tauc variable as a function of energy can be observed, as well as the linear performance due to electronic transitions from the valence band to the conduction band. Transmittance decrease when the CdS nanoparticles are included due to greater amounts of material having dispersed on the surface, as observed in Figure 3, lines B1 and B2 and in Figure 4 lines D1 and D2. The decrease in the transmittance indicates that there is better homogeneity and lower roughness in the thin film [8,13]. When thermal treatment is applied the material becomes denser and transmittance is reduced [40]. In thin films with both one and two layers, absorption edges are traversed at longer wavelengths and the thin films absorption is improved when nanoparticles are incorporated [31]. This behavior is attributed to the quantum confinement and the arrangement of the nanoparticles deposited on a surface of the same chemical nature, which leads to a reduction of the band gap [6,32].

Figure 6a,b show the effects of CdS nanoparticle inclusion and thermal treatment on thin film band gaps. As can be seen, the band gap decreases when thermal treatment is applied [50]. Furthermore, the band gap decreases when CdS nanoparticles are added [5]. The same behavior is observed when the thermal treatment is performed after the incorporation of nanoparticles. Figure 6b shows the same trend in the band gap behavior of CdS thin films with two layers. However, the values in Figure 6a tend to be smaller than those in Figure 6b. The band gap decreases with the increasing grain size, resulting in a red shift on the optical absorption edge, as shown in the transmittance behavior. This is similar to the transmittance behaviour of a semiconductor and represents a reasonable level of electrical conductivity [5,6].

In the present work, we report on the preparation of homogeneous and compact microstructured CdS films. In the next section, we focus on CdS films deposited with one and two layers annealing with and without CdS nanoparticles. 

### 3.2. X-ray Diffraction (XRD) Analysis

Figure 7a shows the diffractograms obtained from samples B1 and B2 and Figure 7b displays the diffractograms obtained from samples D1 and D2. As can be seen, annealed samples presented structural arrangement [18,27,49] showing a diffraction pattern at 2θ (degree) = 26.5 indexed to (002) or (111). It is related to the characteristics of a hexagonal structure [17,27] or a cubic crystalline structure of CdS [25,48] respectively. Furthermore, these results are consistent with those of other authors [51,52]. Cubic and hexagonal structures have been previously reported for optoelectronic applications and in general in semiconductor devices [31,44,50,52,53]. 

We reported the hexagonal structure for CdS thin films without thermal treatment for the recipe reported in a previous work [40] and for CdS NpS as synthesized in Carrillo-Castillo et al. [46] (Appendix A). The XRD pattern of the CdS film deposited by CBD in two layers without annealing shows a preferential growth in the (002) direction of the hexagonal crystalline phase and does not show clear diffraction for other orientations [53,54] (Appendix A). 

Appendix A shows the XRD pattern of CdS NpS from previous work. As we reported, several peaks of hexagonal phase are shown due to diffraction from (100), (002), (111), (101), (102), (220), (103), (112) planes of CdS [39,55,56,57].

In addition, other authors reported hexagonal crystalline structure for CdS films deposited via soft chemistry methods with control of chemical and physical parameters without using high temperatures in the synthesis or thermal treatment in deposited films [51,53]. 

According to the results, the described homojunction can be obtained from CdS/CdS NpS since part of CdS layer deposited by CBD still acts as host material surface of CdS NpS and the crystalline structure reported by CdS deposited by CBD and CdS NpS is not affected when we create a second layer of these materials. However, the band gap and electrical conductivity are affected by the formation of double layers [58]. The mean value of the crystallite size was determined using the Sherrer equation [59]. The main plane at 26.4° was chosen to calculate the crystallite sizes. A Lorentz model was used to fit the full width at half maximum (FWHM). The corresponding sizes of the crystallites are 28.4 nm and 21.3 nm for B2 and D2, respectively.

### 3.3. Raman Spectroscopy Analysis 

Figure 8 shows the Raman spectra of the films used to analyze the influence of CdS nanoparticle addition and thermal annealing on the CdS films.

These Raman dispersion peaks at 303 and 610 cm^−^^1^ are characteristic of CdS longitudinal optical phonons (LO) [6,7,40], referred to as 1LO (first order) and 2LO (second order), respectively. The Raman spectra of all films showed two main peaks: 1-longitudinal optical (1LO) phonon mode at 303 cm^−^^1^ and 2LO mode at 610 cm^−^^1^. The intensity for both signals increases as the number of CdS layers is increased and the annealing treatment is applied. This effect is related to the higher formation of dense crystalline films. Figure 8b shows the Raman spectra of samples D1 and D2. It shows an intensity at the dispersion peak of 610 cm−1, which represents the 2LO (second order) [22,31]. 

The Raman spectra for CdS films deposited via CBD with one and two layers annealed, respectively, show the LO frequencies shift from 303 cm^−^^1^ to 300 cm^−^^1^, and 610 cm^−^^1^ to 559 cm^−^^1^. This could be due to the surface optical phonon (SOP) mode effect, where SOP modes are observed for smaller particle sizes and the wavelength of the excitation laser light inside the particles [60,61,62]. In the Scanning electron microscopy (SEM) section it can be confirmed that s CdS films deposited by CBD have smaller particles than CdS films/CdS NpS on the surface.

Literature for the CdS crystal has reported the 1-LO phonon mode at 305 cm^−^^1^ whose intensity is due to a better order in the crystallinity of the CdS thin film [54,55]. This behavior is observed in B1 vs. B2 and D1 vs. D2 due to the structural arrangement shown in the X-ray diffraction analysis (see Figure 7).

### 3.4. Scanning Electron Microscopy (SEM) Analysis 

Figure 9 shows SEM images of the thin films. Figure 9c,f show the homogeneous morphology at the surface of CdS thin films deposited via CBD and annealing. There is an increase in the cluster size as the number of layers increases in CdS films deposited via CBD when thermal treatment is applied [33]. As can be seen in Figure 9a, the sample B1 shows that the nanoparticles are homogeneously dispersed on the surface of the film. Figure 9b shows the same thin film with thermal treatment (sample B2). It can be seen that these nanoparticles are embedded due to the diffusion caused by this treatment, and for this reason, the roughness achieved was lower in this film [28]. In both films (B1 and B2) nanocrystalline grains can be seen in a compact granular structure with very well-defined grain boundaries; these are the characteristics of the ion-by-ion growth mechanism of the chemical bath deposition process [6,13]. 

The dispersion of CdS nanoparticles in sample D1 (Figure 9d) was similar to that in sample B1. However, slightly more agglomerated material and more clusters were observed [13]. Figure 9e shows the SEM image of sample D2. The same thin film shows a fracture in the surface of the film after thermal treatment. This could be attributed to the fact that the film was stressed by the excess of material. Thus, the roughness of this film was the highest of the four films examined. The mean particle size (MND) was determined using FIJI ImageJ image analysis software; the estimated size is shown in each image. 

### 3.5. Atomic Force Microscopy (AFM) Analysis 

Figure 10a shows the atomic force microscope image of sample B1. The image obtained shows a surface roughness of 3.3 nm and an average roughness of 2.4 nm. On the other hand, Figure 10b shows the image obtained from sample B2, where a surface roughness of 2.7 nm and an average roughness of 2.17 nm was observed. For these samples, it is observed that a decrease of roughness when applying the thermal treatment may lead to slight annealing effects, resulting in the smoother surface. 

Figure 11a shows the image of sample D1, which obtained a surface roughness (RMS) of 4.09 nm and an average roughness of 3.20 nm; these values increased compared to those in Figure 10a due to the addition of another layer of material. In Figure 11b, sample D2 obtained a surface roughness (RMS) of 4.77 nm and an average roughness of 3.77 nm. 

The roughness values of the CdS films are less than 5 nm, indicating that the surface of the thin CdS films is relatively compact, uniform and highly dense with good homogeneity (more peaks than valleys) and homogeneous height distribution [13,52].

### 3.6. Electrical Conductivity

The electrical conductivity of CdS films with and without CdS nanoparticles was obtained. Figure 12a,b show the current (I)–voltage (V) characteristics of samples A2 and C2. The I-V characteristics do not show an important dependence on the number of layers of films. However, the I–V curves show spectral characteristics, a linear behavior of the current as a function of voltage. Figure 12c shows the I-V characteristics of samples B1 and B2. As can be seen, the electrical conductivity of the thin films was improved by the addition of CdS nanoparticles. This fact could be attributed to the structural changes due to the inclusion of CdS nanoparticles. The dispersion of CdS nanoparticles leads to improved structural properties due to the increase in the average grain size, which was confirmed by SEM and XRD measurements. Thus, the grain boundary area is reduced, which results in stronger electrical conductivity. Figure 12d shows the I–V curves of samples D1 and D2. As can be observed, the electrical conductivity is higher than that measured in the thin films without CdS nanoparticles. In both cases, the conductivity decreases with the thermal treatment; however, they remain within acceptable values and maintain good electrical conductivity behavior [40]. The results demonstrate the potential use of these thin films in the development of semiconductor devices. 

## 4. Conclusions

In this study, we report the construction of CdS homojunctions based on CdS thin films deposited by CBD and CdS nanoparticles synthesized using the precipitation method. CdS films of one and two layers via CBD on glass substrates with CdS nanoparticles were successfully prepared using the spin coating method. In particular, the influence of film composition and processing on optical, chemical, morphological, and electrical properties was studied. Raman spectra of the films showed characteristic peaks at 303 and 610 cm^−^^1^, corresponding to the first and second order longitudinal phonons of the CdS. UV-Vis results of the modified films with CdS nanoparticles showed higher wave shifts due to the nanoparticles improving their absorption and decreasing the band gap. AFM measurements confirmed the good homogeneity and the low roughness of the films, while SEM images revealed that CdS nanoparticles were deposited on the surfaces of the thin films, and the thermal treatment applied densified the material, rearranging its crystalline structure, as confirmed by the diffractograms appreciated by XRD. In summary, the incorporation of CdS nanoparticles to the metal-CdS represents an adequate approach to developing semiconductor layers for different optoelectronic applications.

## Figures and Tables

**Figure 1 micromachines-14-01168-f001:**
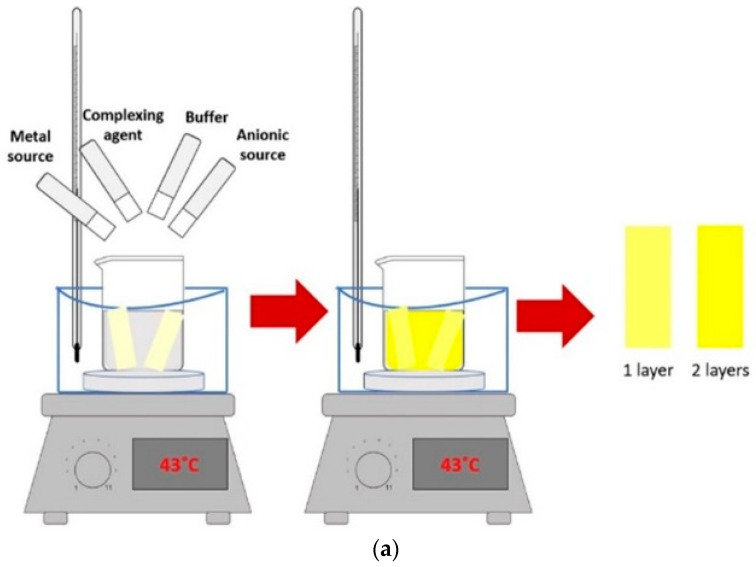
(**a**) Schematic illustration of chemical bath deposition of CdS thin films. (**b**,**c**) photographs of CdS thin films deposited in one and two layers; A1 (1 CBD CdS as-deposited) and C1 (2 CBD CdS as-deposited) respectively. As-deposited: thin films without annealing.

**Figure 2 micromachines-14-01168-f002:**
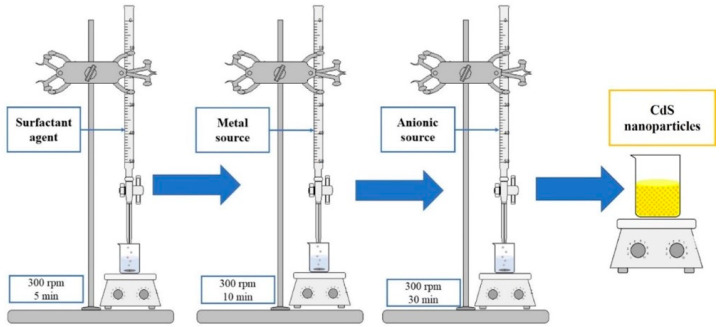
Solution method for the synthesis of CdS nanoparticles.

**Figure 3 micromachines-14-01168-f003:**
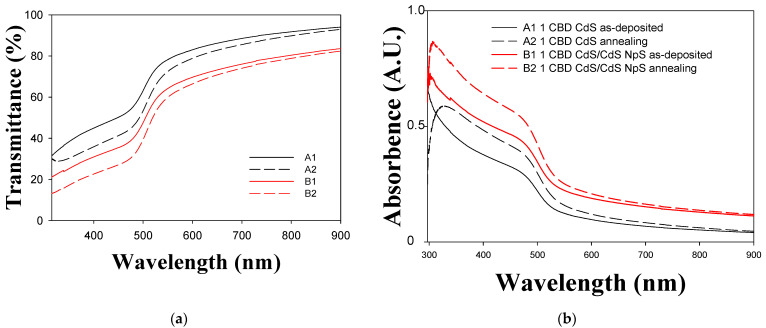
UV-Vis transmittance and absorbance spectra of CdS thin films (1 layer). (**a**) Transmittance; (**b**) absorbance.

**Figure 4 micromachines-14-01168-f004:**
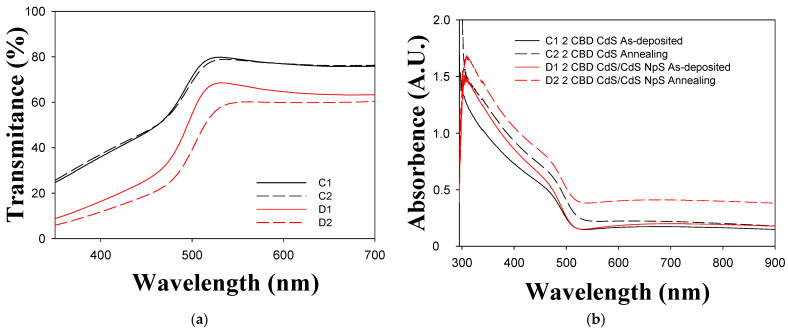
UV-Vis transmittance and absorbance spectra of CdS thin films (2 layers). (**a**) Transmittance; (**b**) absorbance.

**Figure 5 micromachines-14-01168-f005:**
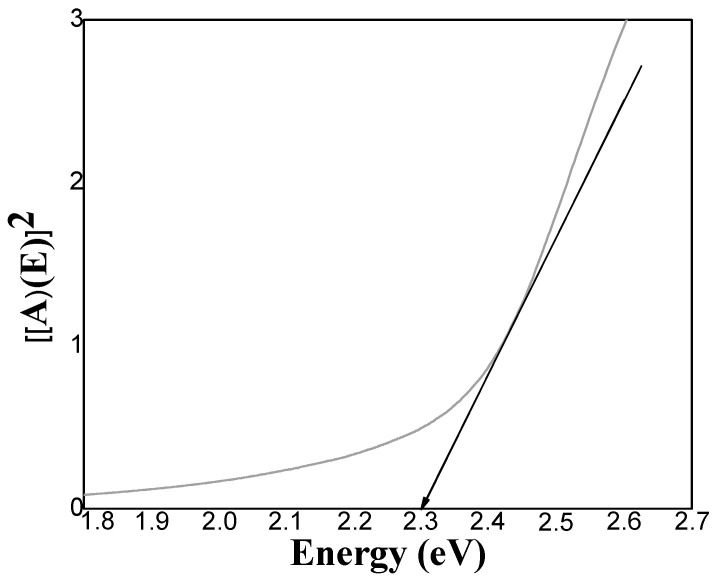
Tauc variable versus Energy plot for selected CdS film; Sample A1 (1 CBD CdS as-deposited).

**Figure 6 micromachines-14-01168-f006:**
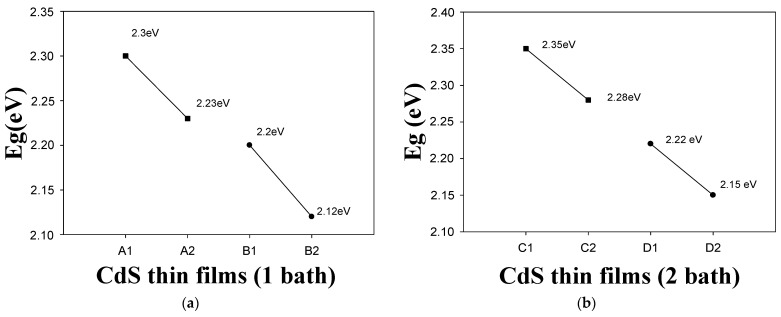
Band gap effects in (**a**) CdS thin films of 1 layer and (**b**) CdS thin films of 2 layers.

**Figure 7 micromachines-14-01168-f007:**
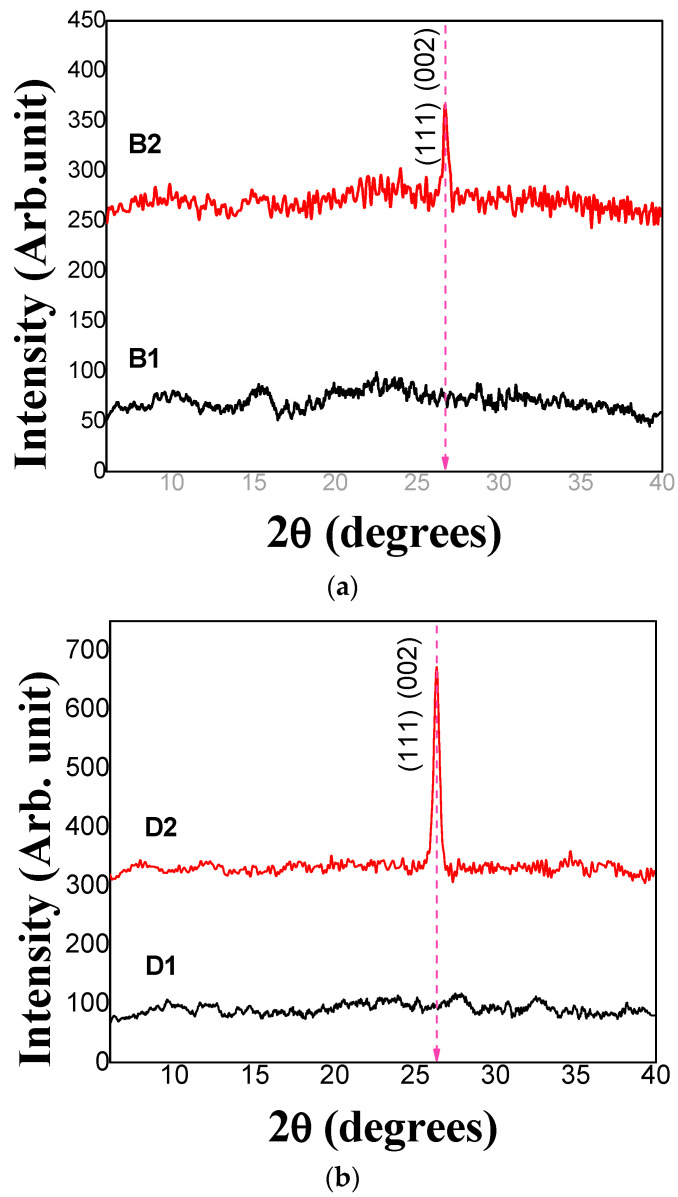
X-ray diffraction pattern of (**a**) sample B1 (1 CBD CdS/CdS NpS as-deposited) and sample B2 (1 CBD CdS/CdS NpS annealing) and (**b**) sample D1 (2 CBD CdS/CdS NpS as-deposited) and sample D2 (2 CBD CdS/CdS NpS annealing).

**Figure 8 micromachines-14-01168-f008:**
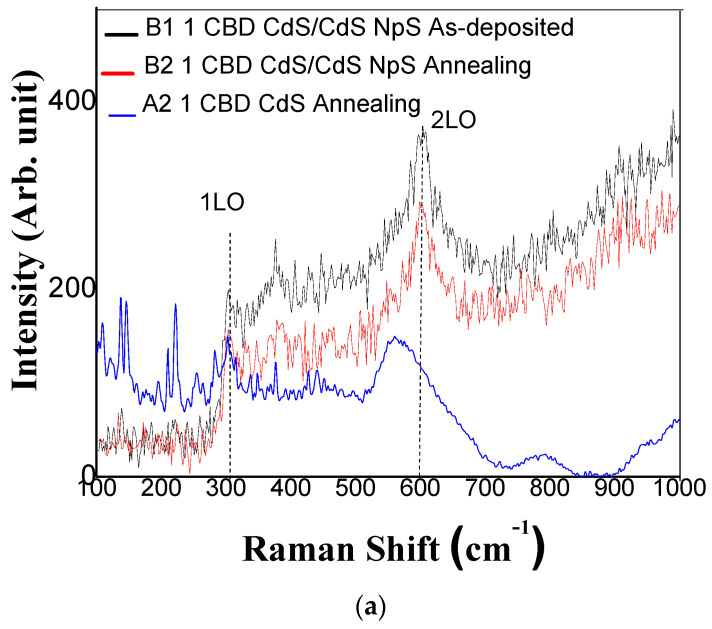
Raman spectra of (**a**) CdS thin films 1 layer and (**b**) CdS thins films 2 layers.

**Figure 9 micromachines-14-01168-f009:**
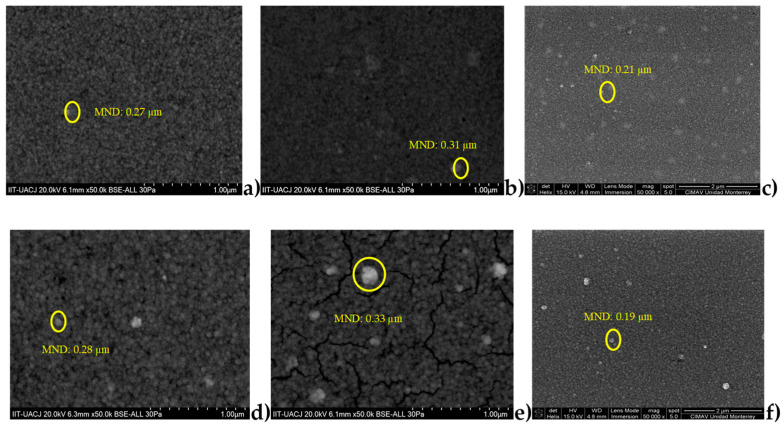
SEM images analyzed with the open-source software FIJI ImageJ. (**a**) sample B1 ( 1 CBD CdS/CdS NpS as-deposited), (**b**) sample B2 ( 1 CBD CdS/CdS Nps annealing), (**c**) sample A2 (1 CBD CdS annealing) and (**d**) sample D1 (2 CBD CdS/CdS NpS as-deposited), (**e**) sample D2 (2 CBD CdS/CdS NpS annealing and (**f**) sample C2 ( 2 CBD CdS annealing).

**Figure 10 micromachines-14-01168-f010:**
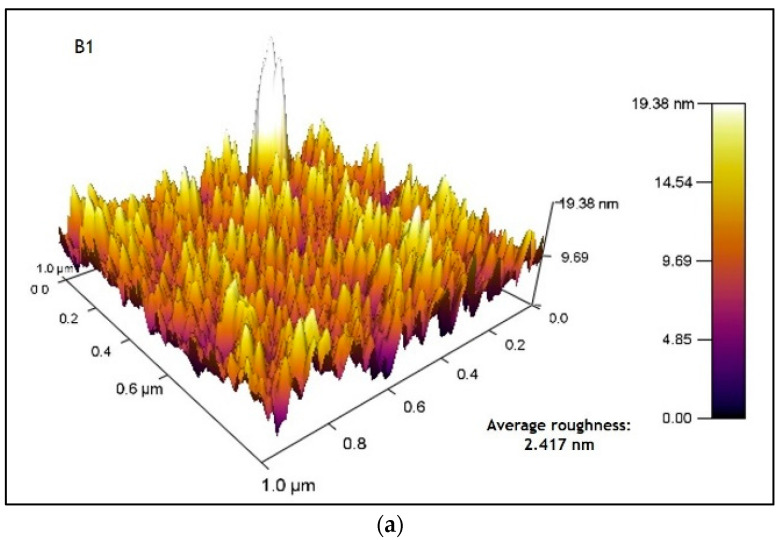
AFM images (**a**) sample B1 (1 CBD CdS/CdS NpS as-deposited) and (**b**) sample B2 (1 CBD CdS/CdS NpS annealing).

**Figure 11 micromachines-14-01168-f011:**
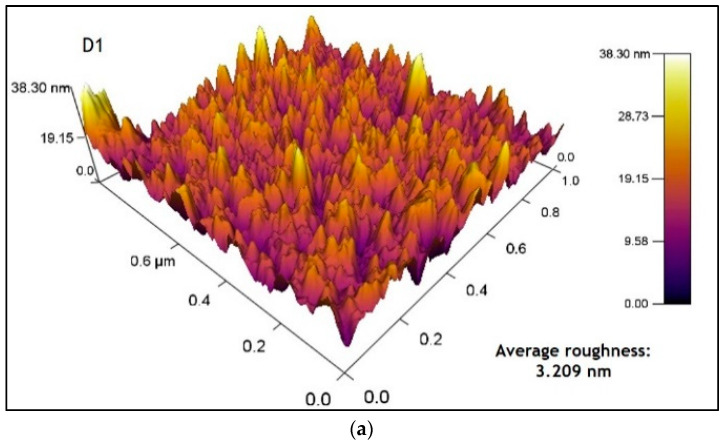
AFM images (**a**) sample D1 (2 CBD CdS/CdS NpS as-deposited) and (**b**) sample D2 (2 CBD CdS/CdS NpS annealing).

**Figure 12 micromachines-14-01168-f012:**
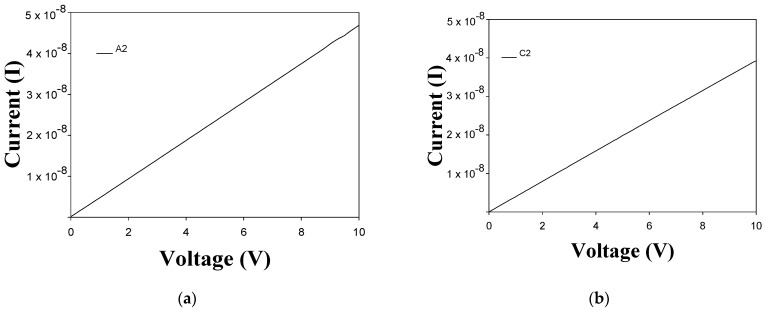
I-V curves of (**a**) sample A2 (1 CBD CdS annealing), (**b**) sample C2 (2 CBD CdS annealing), (**c**) samples B1 (1 CBD CdS/CdS NpS as-deposited) and sample B2 (1 CBD CdS/CdS NpS annealing), (**d**) sample D1 (2 CBD CdS/CdS NpS as-deposited) and sample D2 (2 CBD CdS/CdS NpS annealing).

**Table 1 micromachines-14-01168-t001:** Films studied as a function of composition and processing.

Name of Samples	Description	Name of Samples	Description
**A1**	1 CBD CdS as-deposited	**C1**	2 CBD CdS as-deposited
**A2**	1 CBD CdS annealing	**C2**	2 CBD CdS annealing
**B1**	1 CBD CdS/CdS NpS as-deposited	**D1**	2 CBD CdS/CdS NpS as-deposited
**B2**	1 CBD CdS/CdS NpS annealing	**D2**	2 CBD CdS/CdS NpS annealing

## Data Availability

Not applicable.

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
