# Peer review of "Study of CdS/CdS Nanoparticles Thin Films Deposited by Soft Chemistry for Optoelectronic Applications"

_micromachines, 2023, doi:10.3390/mi14061168_

Round 1
Reviewer 1 Report
The authors have presented the investigation on the fabrication, characterization and optoelectronic application of homojunction based on CdS/CdS NpS using soft chemistry methods in which CdS thin films were prepared by chemical bath deposition, while CdS nanoparticles were obtain via precipitation method. The manuscript is well presented with sufficient data and the materials were thoroughly characterized. However, there are certain shortcomings which must be addressed before the acceptance of the article. Such as,
1. The authors should briefly highlight the importance of soft chemistry techniques with supported references
2. Paragraphs are haphazardly arranged in the introduction which should be corrected
3. There should not be a space in the word CdS / CdS, thus correct it elsewhere in the article as well
4. There should be space between the number and the unit of temperature, (70˚C, should be 70 ˚C).
5. In figure 9, the internal axis should be included
6. The manuscript is well written, however, at several places various typographical and grammatical mistakes have been found. Such as, in page 1, line 23 and many more like this, thus the manuscript should be thoroughly checked for these errors
see the report
Author Response
Response to referees for Ms. Ref. No.: Manuscript ID: micromachines-2411821
Dear Reviewers: We sincerely thank the reviewers for the time invested in reviewing our paper as well as their valuable comments. We have revised the paper taking into account the comments provided. We are now submitting the revised manuscript for review and potential publication.
Thank you for the comments and improvements made to the article. We have responded point by point to each comment as indicated below and improved the English language and style.
Response to reviewer #1
1. The authors should briefly highlight the importance of soft chemistry techniques with supported references.
R. The authors have add highlight about the importance of soft chemistry techniques in the introduction with supported references.
“Every deposition method has its advantages and limitations; however, the soft chemistry techniques have been studied extensively due to simplicity, low deposition temperatures, low cost, large area deposition, and low risk synthesis [2,6,8,10,12,19-21,25-26]. Two major important benefits of the soft chemistry techniques can be considered: preparation of new metastable phase and preparation of nanosize materials.”
2. Paragraphs are haphazardly arranged in the introduction which should be corrected
R. The authors have organized some paragraphs and have used connectors between some sentences for a better understanding. In addition, the authors have edited extensively English language, style and revised typos in manuscript.
3. There should not be a space in the word CdS / CdS, thus correct it elsewhere in the article as well
R. The authors have been corrected in all manuscript eliminated space in the word CdS/CdS according to reviewer comment.
4. There should be space between the number and the unit of temperature, (70˚C, should be 70 ˚C).
R. The authors have been corrected 70˚C for 70 °C.
“The drying step was performed after the CdS nanoparticles deposition at a temperature of 70 ˚C for 10 min in”
5. In figure 9, the internal axis should be included
R, The Authors have added internal axis.
See Figure 8. Raman spectra of a) CdS thin films 1 layer and b) CdS thins films 2 layers
6. The manuscript is well written, however, at several places various typographical and grammatical mistakes have been found. Such as, in page 1, line 23 and many more like this, thus the manuscript should be thoroughly checked for these errors
R. The authors have edited extensively English language, style and revised typos in manuscript.

Reviewer 2 Report
Study of CdS / CdS nanoparticles thin films deposited by soft chemistry for optoelectronic applications
In the present manuscript, the authors studied cadmium sulphide (CdS) thin films containing nanoparticles of the same material as the active layer. The main aim of the work is its application in fabricating optoelectronic devices. The authors obtained CdS thin films and CdS nanoparticles via a low-temperature soft chemistry technique. The CdS thin film was deposited by chemical bath deposition (CBD); the CdS nanoparticles were synthesized by the precipitation method. The author characterizes the synthesised samples using XRD, UV-Vis spectrophotometer, and Raman spectroscopy and SEM, AFM techniques.
The present manuscript looks fine, except for a few points that must be clearly stated in the manuscript:
- For a good presentation of the manuscript, General comment, for all figures presented in the manuscript, they need to be very clear and of comparable size for the axis caption numbers below the axis’s and many more.
- In the XRD data in Figs. 7 and 8, what is the source of the background intensity, and why did the authors leave the background intensity as it is without removal?
- Also, the authors perform very up-to-date analysis using up-to-date techniques, however, I can’t see a scientific and logical correlation between results produced from XRD, UV-Vis STM, and AFM techniques, in particular with the samples containing CdS nanoparticles.
Ø Finally, fine tuning of the English is required
With my best
Study of CdS / CdS nanoparticles thin films deposited by soft chemistry for optoelectronic applications
In the present manuscript, the authors studied cadmium sulphide (CdS) thin films containing nanoparticles of the same material as the active layer. The main aim of the work is its application in fabricating optoelectronic devices. The authors obtained CdS thin films and CdS nanoparticles via a low-temperature soft chemistry technique. The CdS thin film was deposited by chemical bath deposition (CBD); the CdS nanoparticles were synthesized by the precipitation method. The author characterizes the synthesised samples using XRD, UV-Vis spectrophotometer, and Raman spectroscopy and SEM, AFM techniques.
The present manuscript looks fine, except for a few points that must be clearly stated in the manuscript:
- For a good presentation of the manuscript, General comment, for all figures presented in the manuscript, they need to be very clear and of comparable size for the axis caption numbers below the axis’s and many more.
- In the XRD data in Figs. 7 and 8, what is the source of the background intensity, and why did the authors leave the background intensity as it is without removal?
- Also, the authors perform very up-to-date analysis using up-to-date techniques, however, I can’t see a scientific and logical correlation between results produced from XRD, UV-Vis STM, and AFM techniques, in particular with the samples containing CdS nanoparticles.
Ø Finally, fine tuning of the English is required
With my best
Author Response
Response to referees for Ms. Ref. No.: Manuscript ID: micromachines-2411821
Dear Reviewers: We sincerely thank the reviewers for the time invested in reviewing our paper as well as their valuable comments. We have revised the paper taking into account the comments provided. We are now submitting the revised manuscript for review and potential publication.
Thank you for the comments and improvements made to the article. We have responded point by point to each comment as indicated below and improved the English language and style.
Response to reviewer #2
1. For a good presentation of the manuscript, General comment, for all figures presented in the manuscript, they need to be very clear and of comparable size for the axis caption numbers below the axis’s and many more.
R. All figures that contain graphics have been edited by authors at same format.
2. In the XRD data in Figs. 7 and 8, what is the source of the background intensity, and why did the authors leave the background intensity as it is without removal?
R. XRD diffraction patterns of CdS samples displayed a broad peak characteristic of amorphous nature of glass substrate.
The authors have corrected the XRD patterns; because the broad hump observed in the 2Ө range of 20–35 2Ө (degrees) corresponded to glass substrate. It is clear the change of crystalline structure for as deposited films vs annealed films. The authors have clarified the effect of annealing treatment on the films.
See Figure 7. X-ray diffraction pattern of a) sample B1 (1 CBD CdS/CdS NpS as-deposited) and sample B2 (1 CBD CdS/CdS NpS annealing) and b) samples D1 (2 CBD CdS/CdS NpS as-deposited) and sample D2 ( 2 CBD CdS annealing).
3. Also, the authors perform very up-to-date analysis using up-to-date techniques, however, I can’t see a scientific and logical correlation between results produced from XRD, UV-Vis STM, and AFM techniques, in particular with the samples containing CdS nanoparticles.
R. In different results sections is explained the final properties studied in each characterization technique and demonstrated the correlation particularly when CdS nanoparticles are deposited on CdS films:
“Transmittance decrease when the CdS nanoparticles are included due to more material dispersed on the surface, as observed in Figure 3, lines B1 and B2 and Figure 4 lines D1 and D2. The decrease in the transmittance indicates that there is a better homogeneity and a lower roughness in the thin film [8,13]. When thermal treatment is applied the material becomes denser and transmittance is reduced [38]. In one and two layers, absorption edg-es are traversed at longer wavelengths and the thin films absorption is improved when nanoparticles are incorporated [29]. This behavior is attributed to the quantum confine-ment and the arrangement of the nanoparticles deposited on a surface of same chemical nature, which leads to a reduction of the band gap [6, 30].
“Figure 6 shows the effect of CdS nanoparticles inclusion and thermal treatment on thin films bandgap. As seen, the band gap decreases when thermal treatment is applied [48]. Besides, the bandgap decreases when CdS nanoparticles are added [5]. The same be-havior is observed when the thermal treatment is performed after the incorporation of na-noparticles. Figure 6b shows the same trend in the band gap behavior of CdS thin films with two layers. However, the values in Figure 6a tend to be smaller than those in Figure 6b. The band gap decreases with increasing grain size, resulting in a red shift on the opti-cal absorption edge, as show in the transmittance behavior. This is more similar to that of a semiconductor and represents a reasonable level of electrical conductivity [5,6].”
“According to the results, described homojunction can be obtained from CdS/CdS NpS since part of CdS layer deposited by CBD still act as host material surface of CdS NpS and the crystalline structure reported by CdS deposited by CBD and CdS NpS is not affected when we create two layers of these materials. However, the band gap and electrical con-ductivity are affected by the formation of double layers [56]. The mean value of the crystal-lite size was determined using the Sherrer equation [57]. The main plane at 26.4° was chosen to calculate the crystallite sizes. A Lorentz model was used to fit the FWHM. The corresponding sizes of the crystallites are 28.4 nm and 21.3 nm for B2 and D2, respective-ly.”
“Figure 8 shows the Raman spectra of the films to analyze the influence of CdS nano-particles addition and thermal annealing on the CdS films.”
“The Raman spectra for CdS films deposited by CBD with one and two layers anneal-ing show the LO frequency shift from 303 cm-1 to 300 cm-1, and 610 cm-1 to 559 cm-1. This could be due to the surface optical phonon (SOP) mode effect, where SOP modes are ob-served for smaller particle sizes and the wavelength of the excitation laser light inside the particles [58–60]. SEM characterization shows that CdS films deposited by CBD (Figures 10a and 10d) have smaller particles than CdS films/CdS NpS on the surface.”
“Literature for the CdS crystal has reported the 1-LO phonon mode at 305 cm-1 whose intensity is due to a better order in the crystallinity of the CdS thin film [54,55]. This be-havior is observed in B1 vs B2 and D1 vs D2 due to the structural arrangement shown in X-ray diffraction analysis (see Figure 7).”
“There is an increase in the cluster size as the number of layers increases in CdS films de-posited by CBD and when thermal treatment is applied [31]. As can be seen in Figure 9b, the sample B1 shows that the nanoparticles are homogeneously dispersed in the surface of the film. Figure 9c shows the same thin film with thermal treatment (sample B2). It can be seen how these nanoparticles are embedded due to the diffusion obtained with this treatment, and for this reason, the roughness achieved was lower in this film [28]. In both films (B1 and B2) nanocrystalline grains can be seen in a compact granular structure with a very well-defined grain boundaries, these are the characteristics of the ion-by-ion growth mechanism of the chemical bath deposition process [6,13].”
“The dispersion of CdS nanoparticles in sample D1 (Figure 9e) was similar to that in the sample B1. However, slightly more agglomerated material and more clusters were ap-preciated [13]. Figure 9f shows the SEM image of sample D2. The same thin film after thermal treatment shows a fracture in the surface of the film. This could be attributed to the fact that the film was stressed by excess of material.”
“Figure 11a shows the image of sample D1, which obtained a surface roughness (RMS) of 4.09 nm and an average roughness of 3.20 nm; these values increased compared to those in Figure 10a due to the addition of another layer of material. In Figure 11b, sample D2 obtained a surface roughness (RMS) of 4.77 nm and an average roughness of 3.77 nm.”
“The roughness values of the CdS films are less than 5 nm, indicating that the surface of the thin CdS films is relatively compact, uniform and highly dense with good homoge-neity, more peaks than valleys and homogeneous height distribution [13].”
“Figure 12c shows the I-V characteristics of samples B1 and B2. As can be seen, the electri-cal conductivity of the thin films was improved by the addition of CdS nanoparticles. This fact could be attributed to the structural changes due to the inclusion of CdS nanoparticles. The dispersion of CdS nanoparticles leads to improved structural properties due to the in-crease in the average grain size, which was confirmed by SEM and XRD measurements. Thus, the grain boundary area is reduced, which results in stronger electrical conductivity. Figure 12d shows the I-V curves of samples D1 and D2. As can be observed, the electrical conductivity is higher than that measured in the thin films without CdS nanoparticles. In both cases, the conductivity decreases with the thermal treatment; however, they remain within acceptable values and maintain a good electrical conductivity behavior [38]. The results demonstrate the potential use of these thin films in the development of semicon-ductor devices.”
4. Finally, fine tuning of the English is required
R. The authors have edited extensively English language, style and revised typos in manuscript.

Reviewer 3 Report
The authors have synthesized bi-layer CdS film/CdS NPs in this manuscript. I could not find the novelty of this work, there are lots of reports on the synthesis. So, I would suggest rejecting it. If authors do some novel measurements then it can be reconsidered. Here are my comments-
1. Authors have stated that Transmittance has increased with the inclusion of CdS nanoparticles. I think the statement is wrong, transmittance should decrease and absorbance should increase.
2. All the figures should have the same format, I mean the axis legends, and inset description etc should be either in Times New Roman format or Arial format.
3. “Figure 5b shows the same trend in the band gap behaviour of CdS thin films with two layers. However, the values in Figure 5a tend to be smaller than those in Figure 5b.” There are no Figures 5a or 5b in the manuscript only Figure 5.
4. Authors are suggested to index the XRD plane in Figures 7a and 7b and also put the unit of 2θ in the x-axis legends.
5. Author claimed that in there are two diffraction peaks at 2θ - 25⁰ and 28⁰, but I could see only one peak after the annealing only. I think it is quite amorphous before annealing and gain its crystallinity after annealing. Please clarify it.
6. Figure 8 needs not to be in the main manuscript, you can put it into the supporting information.
7. Similarly, authors are strongly recommended to index the Raman spectrum in Figure 9.
8. SEM image quality is very poor, authors are suggested to provide high-resolution images. Also suggested arranging the figure according to the write-up in the manuscript serially. And what is “MND” author has to mention it elsewhere in the manuscript.
9. As CdS is a good light-absorbing material, authors are suggested to measure the visible light photodetection property of those films. Otherwise, I could not find the purpose of the synthesis of CdS nanostructures film. There are lots of reports on it in the literature.
1. There are lots of typos, authors are suggested to correct it.
Needs to be improved.
Author Response
Response to referees for Ms. Ref. No.: Manuscript ID: micromachines-2411821
Dear Reviewers: We sincerely thank the reviewers for the time invested in reviewing our paper as well as their valuable comments. We have revised the paper taking into account the comments provided. We are now submitting the revised manuscript for review and potential publication.
Thank you for the comments and improvements made to the article. We have responded point by point to each comment as indicated below and improved the English language and style.
Response to reviewer #3
1. Authors have stated that Transmittance has increased with the inclusion of CdS nanoparticles. I think the statement is wrong, transmittance should decrease and absorbance should increase.
R. Reviewer´s comment is correct. The authors have modified the statement related to discussion in UV-VIS section.
“Transmittance decrease when the CdS nanoparticles are included due to more material dispersed on the surface, as observed in Figure 3, lines B1 and B2 and Figure 4 lines D1 and D2. The decrease in the transmittance indicates that there is a better homogeneity and a lower roughness in the thin film [8,13].”
In addition, we have corrected the labels of Figure 4a.
See Figure 4. UV-Vis transmittance and absorbance spectra of CdS thin films (2 layers).
2. All the figures should have the same format, I mean the axis legends, and inset description etc should be either in Times New Roman format or Arial format.
R. All figures that contain graphics have been edited by authors at same format.
3. “Figure 5b shows the same trend in the band gap behaviour of CdS thin films with two layers. However, the values in Figure 5a tend to be smaller than those in Figure 5b.” There are no Figures 5a or 5b in the manuscript only Figure 5.
R. Reviewer´s comment is correct. The authors have modified the statement related to band gap CdS thin films disscusion with the correct number of Figure 6.
“Figure 6 shows the effect of CdS nanoparticles inclusion and thermal treatment on thin films bandgap. As seen, the band gap decreases when thermal treatment is applied [48]. Besides, the bandgap decreases when CdS nanoparticles are added [5]. The same be-havior is observed when the thermal treatment is performed after the incorporation of na-noparticles. Figure 6b shows the same trend in the band gap behavior of CdS thin films with two layers. However, the values in Figure 6a tend to be smaller than those in Figure 6b.”
4. Authors are suggested to index the XRD plane in Figures 7a and 7b and also put the unit of 2θ in the x-axis legends.
R. The authors have indexed the XRD planes and we put the unit of 2θ in the x-axis legends.
See Figure 7. X-ray diffraction pattern of a) sample B1 (1 CBD CdS/CdS NpS as-deposited) and sample B2 ( 1 CBD CdS/CdS NpS annealing) and b) samples D1 (2 CBD CdS/CdS NpS as-deposited) and sample D2 ( 2 CBD CdS annealing).
5. Author claimed that in there are two diffraction peaks at 2θ - 25⁰ and 28⁰, but I could see only one peak after the annealing only. I think it is quite amorphous before annealing and gain its crystallinity after annealing. Please clarify it.
R. The authors have restructured the statement about XRD discussion, we considerate the reviewer´s comment. The authors have clarified the effect of annealing treatment on the films.
“Figure 7a shows the diffractograms obtained from samples B1 and B2 and Figure 7b displays the diffractograms obtained from samples D1 and D2. As can be seen, annealed samples presented structural arrangement [18, 26, 47] showing a diffraction pattern at 2Ɵ (degree) = 26.5 indexed to (002) or (111).
6. Figure 8 needs not to be in the main manuscript, you can put it into the supporting information.
R. The authors have moved the Figure 8 in Supporting Information as recommended by reviewer.
7. Similarly, authors are strongly recommended to index the Raman spectrum in Figure 9.
R. The authors have indexed the Raman spectrum according to reviewer´s recommendation.
See Figure 8. Raman spectra of a) CdS thin films 1 layer and b) CdS thins films 2 layers.
8. SEM image quality is very poor, authors are suggested to provide high-resolution images. Also suggested arranging the figure according to the write-up in the manuscript serially. And what is “MND” author has to mention it elsewhere in the manuscript.
R. The quality of SEM images was improved and we describe that the particle size (MND) was determinate using FIJI ImageJ software. The images were arranged in the order that have been displayed.
“The particle size (MND) was determinate using FIJI ImageJ image analysis software, the estimated size is shown into each image.”
See Figure 9. SEM images analyzed with the open-source software FIJI ImageJ.
9. As CdS is a good light-absorbing material, authors are suggested to measure the visible light photodetection property of those films. Otherwise, I could not find the purpose of the synthesis of CdS nanostructures film. There are lots of reports on it in the literature.
R. The authors are grateful for the reviewer's comment but unfortunately cannot perform the experimental set proposed and will take it into account for future publications.
10. There are lots of typos, authors are suggested to correct it.
R. The authors have edited extensively English language, style and revised typos in manuscript.

Round 2
Reviewer 3 Report
The authors have performed sufficient modifications to the revised version of the manuscript. Now the manuscript can be considered for the publication in “Micromachines” journal.
Authors are suggested to include some discussion about recent literature in the manuscript “https://doi.org/10.1038/s41598-022-19340-z, https://doi.org/10.1021/acsami.0c16972, https://doi.org/10.1016/j.spmi.2019.03.024”

Author Response
Response to referees for Ms. Ref. No.: Manuscript ID: micromachines-2411821
Dear Reviewers: We sincerely thank the reviewers for the time invested in reviewing our paper as well as their valuable comments. We have revised the paper taking into account the comments provided. We are now submitting the revised manuscript for review and potential publication.
Thank you for the comments and improvements made to the article. We have responded point by point to each comment as indicated below and improved the English language and style.
Response to reviewer #3
1. The authors have performed sufficient modifications to the revised version of the manuscript. Now the manuscript can be considered for the publication in “Micromachines” journal. Authors are suggested to include some discussion about recent literature in the manuscript : https://doi.org/10.1038/s41598-022-19340-z, https://doi.org/10.1021/acsami.0c16972, https://doi.org/10.1016/j.spmi.2019.03.024
R. The authors have included recommended references in the manuscript, it is important to consider that discussions presented by the authors before are also supported by recommendation references from the reviewer.
“For CdS applications mentioned before, CdS can be presented in different materials, mainly studied in thin films and nanoparticles. There are several physical and chemical methods for the deposition of CdS thin films, among the most commonly used are vacuum evaporation [17], pulsed laser ablation (PLA) [18], electrodeposition [19], molecular beam epitaxy (MBE) [17], sputtering, successive ionic layer adsorption and reaction (SILAR) [20], chemical spray pyrolysis (CSP) [21], RF magnetron sputtering [22], hydrothermal synthesis [23], and chemical bath deposition (CBD) [8,13,24,25]. “
25. Ouafi Mouad,; Jaber Boujemaa,; Laanab, Larbi,; Low temperature CBD growth of CdS on flexible substrates: Structural and optical characterization. Matter. Superlattices and Microstructures. 2019, 129, 212-219, doi.org/10.1016/j.spmi.2019.03.024.
“In addition, there are different processes used for the synthesis of nanoparticles, such as hydrothermal or solvothermal techniques [26,27], microwave approaches [30], and chemical vapor, but these treatments are expensive, extensive, toxic and energy-consuming.”
27. Sukdev, Dolai; Pradip Maiti; Arup Ghorai,; Ritamay Bhunia,; Pabitra, Kumar, and Dibyendu Ghosh. Exfoliated Molybdenum Disulfide-Wrapped CdS Nanoparticles as a Nano-Heterojunction for Photo-Electrochemical Water Splitting. ACS Appl. Mater. Interfaces. 2021, 13, 438-448, doi.org/10.1021/acsami.0c16972
“As can be seen, annealed samples presented structural arrangement [18, 27, 49] showing a diffraction pattern at 2Ɵ (degree) = 26.5 indexed to (002) or (111). It is related to the charac-teristic of hexagonal structure [17,27] or cubic crystalline structure of CdS [48,25] respectively. Furthermore, these results are consistent with those of authors [51,52].”
25. Ouafi Mouad,; Jaber Boujemaa,; Laanab, Larbi,; Low temperature CBD growth of CdS on flexible substrates: Structural and optical characterization. Matter. Superlattices and Microstructures. 2019, 129, 212-219, doi.org/10.1016/j.spmi.2019.03.024.
27. Sukdev, Dolai; Pradip Maiti; Arup Ghorai,; Ritamay Bhunia,; Pabitra, Kumar, and Dibyendu Ghosh. Exfoliated Molybdenum Disulfide-Wrapped CdS Nanoparticles as a Nano-Heterojunction for Photo-Electrochemical Water Splitting. ACS Appl. Mater. Interfaces. 2021, 13, 438-448, doi.org/10.1021/acsami.0c16972.
52. Najm, A.; Hasanain S.; Hasan Sh.; Aisha S. An in-depth analysis of nucleation and growth mechanism of CdS thin film synthesized by chemical bath deposition (CBD) technique. Scientific Reports. 2020, 17(11), 537–547, doi.org/10.1038/s41598-022-19340-z.
“The roughness values of the CdS films are less than 5 nm, indicating that the surface of the thin CdS films is relatively compact, uniform and highly dense with good homogeneity, more peaks than valleys and homogeneous height distribution [13,52].
52. Najm, A.; Hasanain S.; Hasan Sh.; Aisha S. An in-depth analysis of nucleation and growth mechanism of CdS thin film synthesized by chemical bath deposition (CBD) technique. Scientific Reports. 2020, 17(11), 537–547, doi.org/10.1038/s41598-022-19340-z.
